# Balance control mechanisms do not benefit from successive stimulation of different sensory systems

Jean-Philippe Cyr[1,2], Noémie Anctil[1,2], Martin Simoneau[1,2]*

**1** Département de kinésiologie, Faculté de médecine, Université Laval, Québec, Québec, Canada, **2** Centre interdisciplinaire de recherche en réadaptation et intégration sociale (CIRRIS) du CIUSSS de la Capitale Nationale, Québec, Québec, Canada

* martin.simoneau@kin.ulaval.ca

**Data Availability Statement:** All relevant data are within the paper and its Supporting Information files.

**Funding:** This research was supported by grant from the Natural Sciences and Engineering

## Abstract

In humans, to reduce deviations from a perfect upright position, information from various sensory cues is combined and continuously weighted based on its reliability. Combining noisy sensory information to produce a coherent and accurate estimate of body sway is a central problem in human balance control. In this study, we first compared the ability of the sensorimotor control mechanisms to deal with altered ankle proprioception or vestibular information (i.e., the single sensory condition). Then, we evaluated whether successive stimulation of difference sensory systems (e.g., Achilles tendon vibration followed by electrical vestibular stimulation, or vice versa) produced a greater alteration of balance control (i.e., the mix sensory condition). Electrical vestibular stimulation (head turned ~90˚) and Achilles tendon vibration induced backward body sways. We calculated the root mean square value of the scalar distance between the center of pressure and the center of gravity as well as the time needed to regain balance (i.e., stabilization time). Furthermore, the peak ground reaction force along the anteroposterior axis, immediately following stimulation offset, was determined to compare the balance destabilization across the different conditions. In single conditions, during vestibular or Achilles tendon vibration, no difference in balance control was observed. When sensory information returned to normal, balance control was worse following Achilles tendon vibration. Compared to that of the single sensory condition, successive stimulation of different sensory systems (i.e., mix conditions) increased stabilization time. Overall, the present results reveal that single and successive sensory stimulation challenges the sensorimotor control mechanisms differently.

## Introduction

Human upright balance is inherently unstable. To reduce the small deviations from a perfect upright body position, information from proprioceptive, vestibular and visual systems are combined [1–4]. These sensory signals are continuously reweighted based on their reliability and specificity to maintain the upright standing position [5–7]. To assess the role of each

Research Council of Canada Discovery program
(RGPIN-2015-04068) to MS. JPC was supported
by scholarships from Fonds de Recherche du
Québec en Santé (FRQ-S) and the Wilbrod-Bhérer
recruitment scholarship. NA received a summer
research scholarship from Laval University, Faculty
of Medicine. The funders had no role in study
design, data collection and analysis, decision to
publish, or preparation of the manuscript.

**Competing interests:** The authors have declared
that no competing interests exist.

sensory cue, it is common to experimentally induce sensory illusions. Electrical vestibular
stimulation (EVS) with the cathode electrode located on the right mastoid and the anode elec-
trode located on the left mastoid induces an increase in the firing rate of the right vestibular
nerve (cathode) and a decrease in in the firing rate of the left vestibular nerve (anode), leading
to a body sway towards the anode [8, 9]. Ankle tendon vibrations also induce body sway result-
ing from the activation of the primary endings of muscle spindles [10]. When vibration is
applied to the Achilles tendon, the firing rates of the gastrocnemius and soleus muscle spindles
increase (i.e., as if these muscles stretched), suggesting a forward body sway. To counterbal-
ance this illusion, the body sways backward [11]. Previous results suggest that vestibular stimu-
lation influences the processing of somatosensory signals [12–14]. Moreover, human
neuroimaging studies have revealed vestibular projections in the primary and secondary
somatosensory cortex [15, 16] and the primary motor cortex and premotor cortex [16–18].
The overlap in brain activation of the vestibular and somatosensory inputs is not simply ana-
tomical but also reflects a functional crossmodal perceptual interaction. Psychophysical studies
have revealed that vestibular stimulation facilitates the detection of cutaneous stimuli, suggest-
ing a vestibular-somatosensory perceptual interaction [12]. Consequently, it is possible that
successive stimulation of these two senses improves balance control.

The first aim of this study was to compare the ability of the sensorimotor mechanisms to
control balance during alterations in vestibular information or ankle proprioception. A mathe-
matical model was used to assess the effects of sensory and motor noise on balance control; the
results indicate that the magnitude of noise in the vestibular system is ~10 times greater than
that of noise in the proprioceptive system [19]. Furthermore, balance control mainly relies on
ankle proprioception [20, 21], and its contribution represents more than 60% of balance con-
trol [22–25]. Thus, we hypothesize that body sway should be larger during Achilles tendon
vibration compared to EVS.

The second aim of this study was to assess whether successive stimulation of different sen-
sory systems alters the performance of the sensorimotor integration mechanisms. During
upright standing, a sudden alteration in one sensory information source normally leads to an
increase in body sway. When a sensory stimulation is repeated over time, however, the ampli-
tude of body sway decreases [4, 26, 27]. During simultaneous sensory stimulation, the attenua-
tion of body sway is limited or it not necessarily transferred to other senses or other muscles
during ankle tendons vibration [26, 28]. It is unclear, however, whether the sensorimotor inte-
gration mechanisms can benefit from successive stimulation of different sensory modalities.
To address the second aim, we compared balance control performance to a condition in which
a single sensory system was altered to a condition in which different sensory systems were
stimulated in a chronological sequence. An increase in body sway following the subsequent
stimulation would suggest that the error signal from different senses alters the performance of
the sensorimotor integration mechanisms.

## Materials and methods

### Participants

Thirty-two participants (16 men, 16 women, age = 23.1 ± 4.5 years, weight = 69.0 ± 13.2 kg,
height = 168.6 ± 10.7 cm) with no known history of neurological or vestibular dysfunction
took part in two separate experiments (16 participants in each experiment). Prior to participa-
tion, participants provided written informed consent. The study was approved by the Biomedi-
cal research Ethics Committee at Laval University (approval number: 2015–119) and
conformed to the Declaration of Helsinki standards.

## Experimental set-up and protocol

Participants stood barefoot on a force platform (model Optima, Advanced Mechanical Technology, Inc., Watertown, MA, USA) with their feet parallel at 10 cm inter-malleoli and their arms alongside. Their eyes were closed, and their head was turned left at approximately 90˚ and extended approximately 18˚. This head orientation parallels the EVS-evoked rotational vector with the horizontal, thus maximizing the balance response to EVS along the anteroposterior (AP) axis [29, 30]. The force platform signals were sampled at 1000 Hz using a 16-bit A/D converter (model NI PCIe-6531, National Instrument, Austin, TX, USA).

During standing, the most common force acting on the body is the ground reaction force (GRF). It has a vertical and two shear components. The shear forces result from anteroposterior (AP) and mediolateral (ML) directions of the body sway [31, 32]. AP GRF indicates a shift in the sway of the body in the AP direction, and is necessary to prevent a fall. Walking on a slippery surface illustrates this, as in the absence of GRF, the foot slides [32]. In this study, the center of pressure (COP) displacements along the ML and AP axes were calculated from the reaction forces and moments of the force platform. All data were filtered using a zero-lag 4th order low-pass Butterworth filter (cut-off frequency 10 Hz). The center of gravity (COG) along the AP axis was estimated using a zero-point-to-zero-point double integration technique [33, 34] with the assumption that the COP coincides with the vertical line passing through the COM when the horizontal ground reaction force is zero.

## Experimental conditions

Applying vibration to a muscle tendon specifically activates the primary endings of the muscle spindle [10, 35]. In this study, vibration (n = 2, Freq.: 70 Hz, amplitude: ~1 mm) was applied to the Achilles tendon to cause a backward body sway, which is known as a vibration-induced postural response [11, 36, 37]. Vestibular stimulation was delivered by applying electrical stimulation to the eighth cranial nerve (i.e., vestibulocochlear). Electrical vestibular stimulation (EVS) activates all the primary afferents of the semicircular canals and otoliths, with a cathodal current increasing the firing rate of the afferent and an anodal current decreasing the firing rate [9, 38]. EVS induces a net equivalent motion vector (EVS vector) based on the vectorial summation of all the activated vestibular afferents [8]. According to this model, EVS applied bilaterally over the mastoid processes (i.e., in a binaural bipolar configuration) results in a net rotation around a vector pointing posteriorly and ~18.8˚ above Reid's plane [39, 40]. Thus, altering the firing rate of the vestibular afferent results in a perceptual illusion of a tilt of falling towards the cathode electrode. To counter this vestibular illusion, muscles are activated creating body sway toward the anode, that is, a backward body sway in the present experiment. To induce vestibular stimulation, a binaural EVS was delivered from a constant-current stimulator (Model DS-5, Digitimer Ltd., Hertfordshire, UK) to electrodes (5 cm$^2$, ValuTrode® X Cloth Neurostimulation Electrodes, Model VTX5050, Axelgaard Manufacturing Co., Ltd., Fallbrook, CA) located over both mastoid processes and stabilized with a headband. The stimulus was a 1 mA current step that lasted 5 s (see the explanation of the conditions below).

Throughout the two experiments, we alternated the sequence of sensory stimulation. In both experiments, there were two conditions: single and mix. Under the single sensory condition, sensory information from one sensory system was altered, whereas under the mix sensory condition, information from the two sensory systems was altered in sequence. Thus, during the first experiment, under the single sensory condition (Fig 1A) Achilles tendon vibration created a backward body sway. This condition served as a basis of comparison with the mix sensory condition. Under the mix condition, information from the vestibular and proprioceptive systems was altered in sequence.

For the second experiment (Fig 1B), different participants were involved. Experiment 2 was like experiment 1, but under the single sensory condition, the vestibular apparatus was stimulated first to create a backward body sway. Then, under the mix sensory condition, the Achilles tendon was vibrated after the vestibular apparatus was stimulated. Trials under the single condition lasted for 20 s and were divided into three different epochs: prestimulation (5 s), stimulation (5 s) and poststimulation (10 s). Under the mix sensory condition, trials lasted for 40 s and contained six epochs: prestimulation (10 s), first stimulation (5 s), post first stimulation (5 s), prestimulation (5 s), second stimulation (5 s), and post second stimulation (10 s). Participants performed, in blocks, 10 trials of each of the two conditions.

## Data analysis

Balance control performance was assessed by calculating the scalar distance between the time series of the COP and COG displacements (Fig 2A) and then calculating the root mean square (RMS) value of the COP-COG scalar distance in each interval. This measure reflects the linear acceleration of the COM, an indicator of balance control performance [41, 42]. We also calculated the stabilization time, which was the time needed to recover balance when sensory information returned to normal (i.e., poststimulation interval). We considered that balance was recovered when the COP-COG scalar distance was below a defined threshold for 500 ms consecutively. For each participant, the threshold was defined as the mean of the RMS value of the COP-COG scalar distance during the prestimulation intervals (dashed line in Fig 2B).

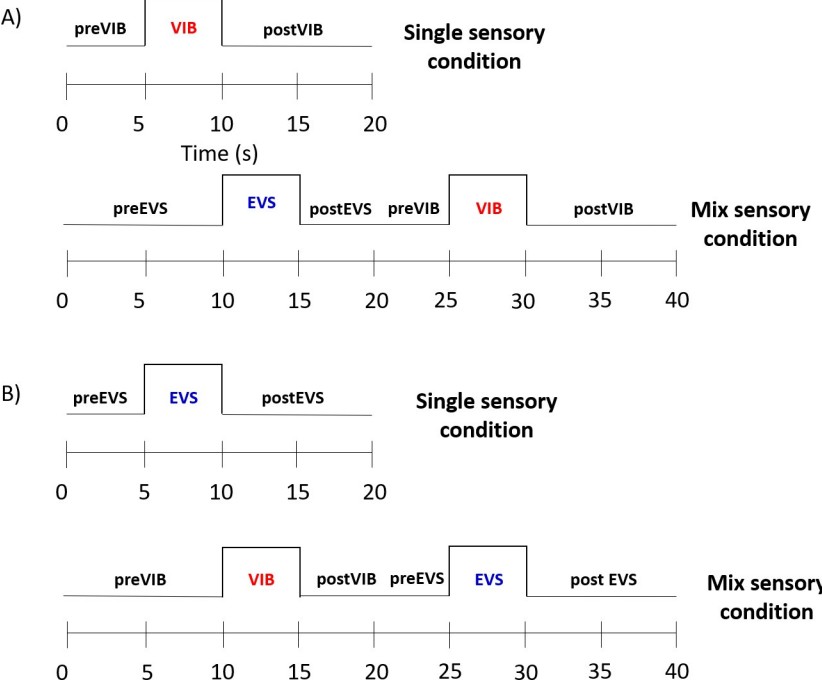

**Fig 1. Temporal sequencing of the epochs for both experiments.** (A) Time course of experiment 1 that contained two experimental conditions: single and mix sensory conditions. Under the single sensory condition, electrical vestibular stimulation (EVS) was applied to induce backward body sway. There were three epochs: preEVS (0–5 s), EVS (5–10 s) and postEVS (10–20 s). Under the mix sensory condition, Achilles tendon vibration (VIB) was applied first to create a backward body sway. Then, EVS was applied. (B) The time course of experiment 2, which also contained two experimental conditions: single and mix sensory conditions. The epoch durations were as in Experiment 1, but under the single sensory condition, the Achilles tendons were vibrated to create a backward body sway. Under the mix sensory condition, first an electrical vestibular stimulation was applied and then the Achilles tendons were vibrated.

Furthermore, we determined the amplitude of balance destabilization immediately following sensory stimulation by identifying the peak force along the AP axis (Fig 2C). Visual inspection of every time series showed that peaks occurred at less than 1.5 s following the stimulation off-set. Comparison of the peaks between conditions allowed us to verify whether balance destabilization differed when vestibular and ankle proprioception returned to normal.

### Statistical analysis

To compare balance control under the single sensory conditions, the RMS values of the COP-COG scalar distance were submitted to analysis of variance (ANOVA) with repeated measures on the factors epoch (prestimulation, stimulation and poststimulation) and condition (VIB, EVS). To contrast balance control performance under the single and mix sensory conditions, the RMS values of the COP-COG scalar distance between the single and mix sensory stimulation were compared through separate ANOVAs with repeated measures on two factors (condition: single, mix; epoch: prestimulation, stimulation and poststimulation). Post-hoc analyses were realized using Tukey's honest significant difference (HSD) test. To assess whether the stabilization time differed between the single and mix conditions, paired t-tests were performed. We evaluated whether balance destabilization (i.e., peak force along the AP axis, following stimulation offset) differed among the single conditions using unpaired T-tests. To compare balance destabilization between the single and mix conditions, paired T-tests were used.

## Results

### Comparison of single sensory conditions

The results of the analysis of the RMS value of the COP-COG scalar distance (Fig 3) during single sensory stimulation partly confirmed our hypothesis, suggesting that balance control

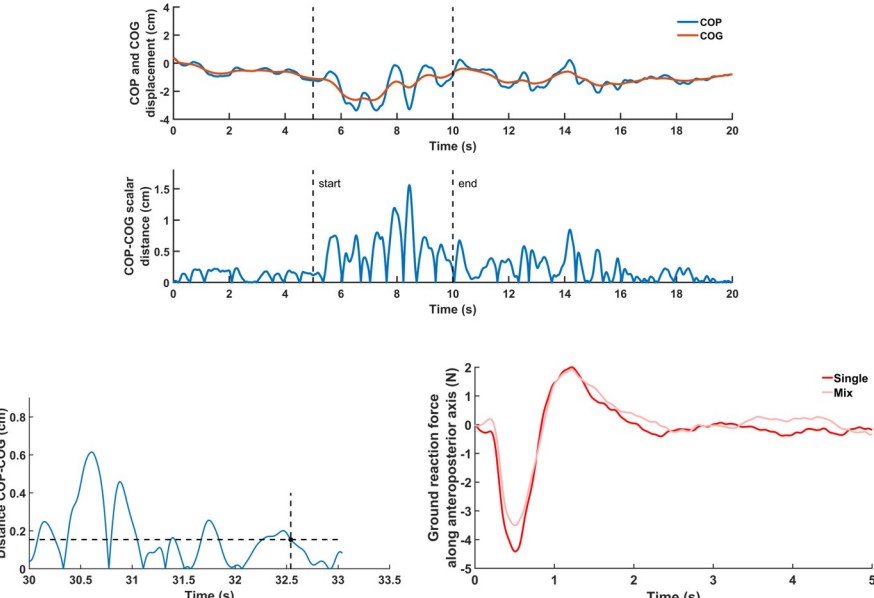

**Fig 2. Illustration of the dependent variables.** (A) Representative time series of the center of pressure (COP) and center of mass (COM) displacements along the anteroposterior axis. Time-series of the corresponding COP-COG scalar difference. (B) Determination of the stabilization time following sensory alteration. The blue line represents the time series of the COP-COG scalar distance, the horizontal black dashed line represents the stability threshold and the crossing vertical black line with the dot depicts the time when the COP-COG scalar distance is below the stability threshold for the next 0.5 s. (C) Mean time-series of the ground reaction force along the anteroposterior axis following Achilles tendon offset for the single and mix conditions.

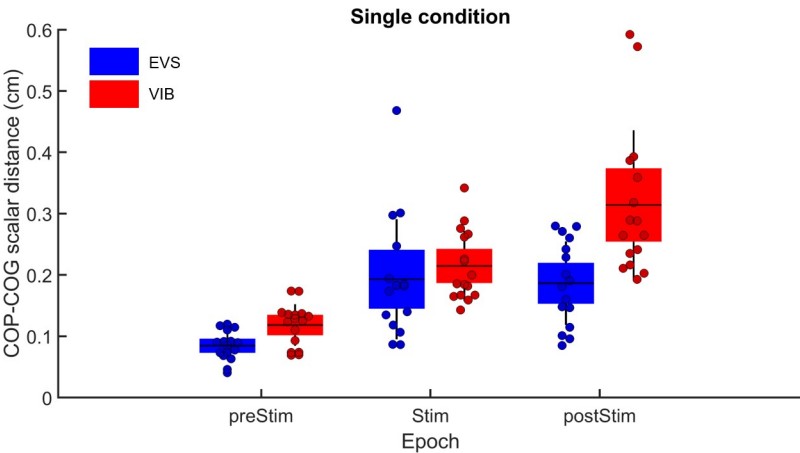

**Fig 3. Single sensory condition: Effect of sensory stimulation on balance control.** Group means of RMS values of the COP-COG scalar distance before (preSTIM), during (STIM) and after (postSTIM) electrical vestibular stimulation and Achilles tendon vibration under the single sensory condition. The dots depict the mean results for each participant. The horizontal lines illustrate the group means, the boxes denote the group standard error of the mean and the lines depict one standard deviation. The blue data and boxes are for the electrical vestibular stimulation (EVS) condition, and the red data and boxes are for the Achilles tendon vibration (VIB) condition.

should be poorer when ankle proprioception is altered compared to EVS (significant interaction of Epoch by Condition: $F_{(2,60)} = 6.82$, $p = 0.002$). Decomposition of the interaction revealed that balance control was similar before and during the alterations in vestibular or ankle proprioceptive signals (ps > 0.05). When sensory information returned to normal, however, as hypothesized, the RMS value of the COP-COG scalar distance was greater following Achilles tendon vibration ($p = 0.009$). Balance control worsened across epochs (main effect of epoch: $F_{(2,60)} = 47.19$, $p = 0.000$). The RMS value of the COP-COG scalar distance increased from the prestimulation to stimulation epochs ($p = 0.0001$) and from the stimulation to post-stimulation epochs ($p = 0.01$). Overall, balance control performance was worse in VIB compared to EVS conditions (main effect of condition: $F_{(1,30)} = 9.78$, $p = 0.004$).

## Balance destabilization under the single sensory condition

Analysis of the peak force along the AP axis revealed that balance destabilization was larger when ankle proprioception returned to normal (mean = −6.24, sd = 22.95) compared to when vestibular information returned to normal (mean = −3.15, sd = 1.52; $t(30) = 3.71$, $p = 0.0008$).

## Comparison of the mix sensory conditions

Comparison of the single to mix conditions when vestibular signals were altered (Fig 4A) revealed worse balance control under the mix compared to single conditions (main effect of condition: $F_{(1,15)} = 6.94$, $p = 0.02$). Furthermore, balance control differed across epochs (main effect of epoch: $F_{(2,30)} = 31.94$, $p = 0.000$). During and following EVS, balance control was similar ($p = 0.98$); however, the RMS values of the COP-COG scalar distance in these epochs were greater than in the preEVS epoch. The analysis reported no difference in balance control performance between conditions across epochs (interaction condition by epoch: $F_{(2,30)} = 1.74$, $p = 0.19$). Comparison of the single to mix sensory conditions when ankle proprioception was altered (Fig 4B) revealed no difference (main effect of condition: $F_{(1,15)} = 0.05$, $p = 0.81$). Across epochs, however, the RMS value of the COP-COG scalar distance varied (main effect of epoch: $F_{(2,30)} = 32.19$, $p = 0.000$). Post-hoc tests showed that the RMS value of

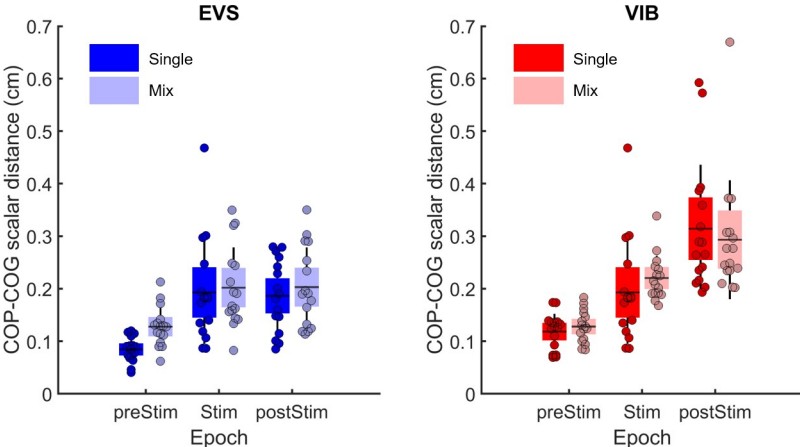

**Fig 4. Mix sensory condition: effect of sensory stimulation on balance control.** (A) Comparison of the RMS values of the COP-COG scalar distance between single sensory and mix sensory conditions, before (preSTIM), during (STIM) and after (postSTIM) electrical vestibular stimulation (EVS). The blue data and boxes represent the single sensory condition, while the light blue data and boxes depict the mix sensory condition. (B) Comparison of the RMS values of the COP-COG scalar distance between the single sensory and mix sensory conditions for the same epochs for the Achilles tendon vibration (VIB) condition. In each panel, the dots depict the mean results for each participant. The horizontal lines illustrate the group means, the boxes denote the group standard error of the mean and the lines depict one standard deviation. The red data and boxes represent the single sensory condition, while the light red data and boxes depict the mix sensory condition.

the COP-COG scalar distance increased across epochs (preStim vs Stim, $p < 0.001$ and Stim vs postStim, $p < 0.01$). Across epochs, however, no difference between conditions was observed (interaction condition by epoch: $F_{(2,30)} = 2.53$, $p = 0.09$).

## Balance destabilization under the mix sensory condition

The peak forces when vestibular information returned to normal were similar between the single (mean = −3.16, sd = 1.52) and mix (mean = −3.47, sd = 1.55; $t_{(15)} = 1.14$, $p = 0.27$) conditions. Following Achilles tendon vibration, the peak force did not differ between the single (mean = −6.24, sd = 2.95) and mix (mean = −5.65, sd = 2.97; $t_{(15)} = −1.84$, $p = 0.09$) conditions.

## Time required to reduce the body sway amplitude

Analysis of the stabilization time between the single and the mix sensory conditions revealed that the stabilization time was longer under mix compared to single sensory conditions when either vestibular or ankle proprioception information returned to normal (Fig 5A and Fig 5B: paired T-tests: $t_{(15)} = −4.29$, $p < 0.001$ and $t_{(15)} = −6.86$, $p < 0.001$, respectively).

The RMS of the scalar distance between the COP and the COG approximates the center of mass acceleration. Thus, the fact that the peak forces immediately following sensory stimulation were alike between the single and mix sensory conditions was not surprising as no difference was observed for the RMS value of the COP-COG scalar distance. Balance destabilization (i.e., peak force) immediately following sensory stimulation was a good predictor of the RMS value of the COP-COG scalar distance; the variance explained by the linear model was larger than 80% (Fig 6). Peak force was, however, not a good predictor of the stabilization time (Fig 7) as the variance ranged from 16% to 51%.

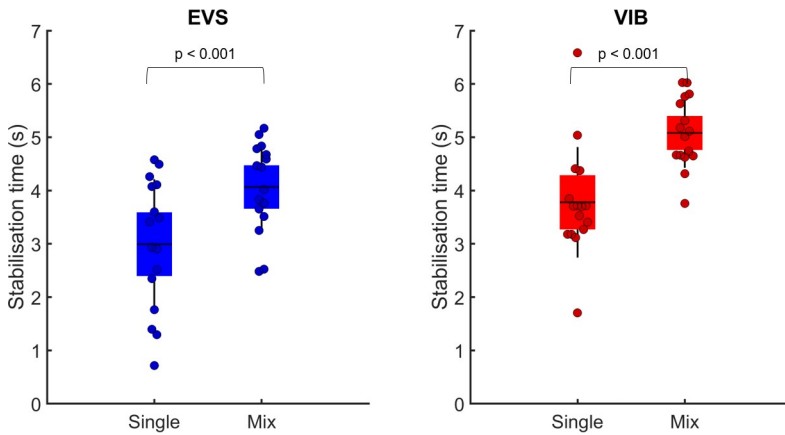

**Fig 5. Effect of single versus mix sensory condition on stabilization time.** (A) Comparison of the stabilization time following electrical vestibular stimulation (EVS) under the single and mix sensory conditions. (B) Comparison of the stabilization time following Achilles tendon vibration (VIB) under the single and mix sensory conditions. In each panel, the dots depict the mean results for each participant. The horizontal lines illustrate the group means, the boxes denote group standard error of the mean and the blue lines depict one standard deviation.

## Discussion

Not much is known about the time course of sensory reintegration following a sudden change in a sensory state. Rapid and effective sensory reweighting is crucial to alter the balance motor commands and to reduce instability. The aim of the present study was two-fold. First, to compare the ability of the sensorimotor mechanisms to reintegrate ankle proprioception and vestibular cues. Second, to investigate whether successive stimulation of different sensory systems altered balance control performance. Contrary to our hypothesis, the results revealed no difference in balance control performance when ankle proprioception or vestibular information was altered. As expected, however, when sensory information returned to normal, balance control performance was poorer during the reintegration of ankle proprioception compared to

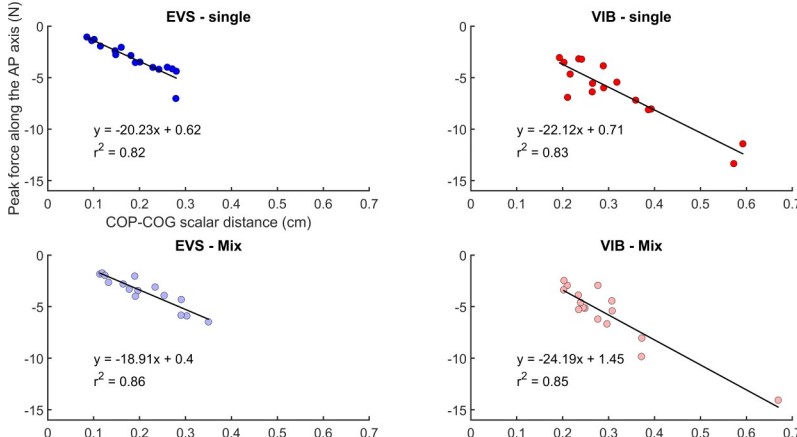

**Fig 6. Relationship between balance destabilization and control when sensory information returned to normal.** Linear relationship between the peak forces and RMS values of the COP-COG scalar distance for the single (upper left panel) and mix sensory conditions (lower left panel) when vestibular information returned to normal (i.e., post stimulation epoch). Linear relationship between peak forces and RMS values of the COP-COG scalar distance for the single (upper left panel) and mix sensory conditions (lower left panel) when ankle proprioception returned to normal (i.e., post stimulation epoch).

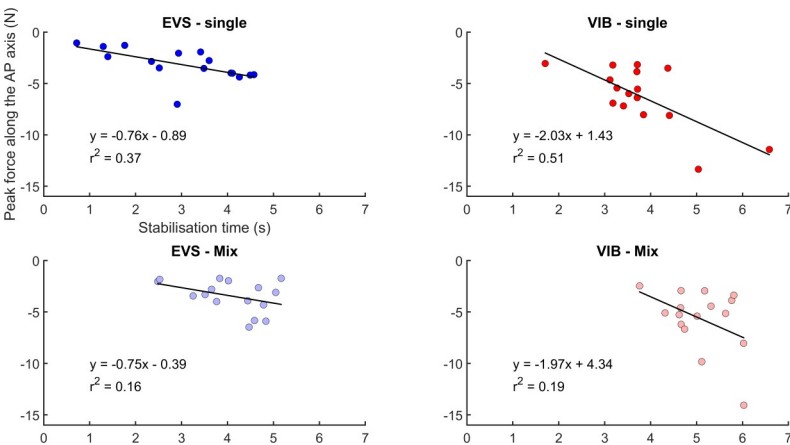

**Fig 7. Relationship between balance destabilization and stabilization time when sensory information returned to normal.** Linear relationship between the peak force and stabilization time for the single (upper left panel) and mix sensory conditions (lower left panel) when vestibular information returned to normal (i.e., poststimulation epoch). Linear relationship between the peak force and stabilization time for the single (upper left panel) and mix sensory conditions (lower left panel) when ankle proprioception returned to normal (i.e., poststimulation epoch).

vestibular information. This difference was caused by a larger peak of the ground reaction force, that is, balance destabilization, following Achilles tendon vibration offset. Following successive changes in the sensory state (i.e., the mix sensory condition), the amplitude of body sway did not differ between the single and mix conditions, but the time needed to recover balance was longer under the mix condition.

## Comparison of the single sensory conditions

Under the single sensory condition, contrary to our hypothesis, balance control performance was similar during ankle proprioception and vestibular alteration. Poorer balance control performance during Achilles tendon stimulation was expected as balance control mainly relies on ankle proprioception [22–25] and the vestibular system is approximately 10 times noisier than the proprioceptive system [19]. We reasoned that during ankle proprioception alterations in the absence of vision, participants would assign larger weights to vestibular information. Since vestibular sensory information is noisy, this should cause poorer balance control. The absence of a difference between ankle proprioception and vestibular alteration may be due to biomechanical constraints. During this stimulation epoch, the amplitude of the backward body sway was restricted by the posterior stability limit. It is tempting to suggest that in the absence of a stability limit, the body sway amplitude could have been larger during Achilles tendon vibration compared to EVS. Furthermore, to reduce balance destabilization during Achilles tendon vibration, it has been suggested that a participant could adopt a forward tilt posture since such a strategy stretches the Achilles tendon and increases ankle stiffness [43]. It is worth noting, however, that the center of mass accelerations (i.e., RMS value of the COP-COG scalar distance) were similar under both conditions, ruling out this latest suggestion. During ankle proprioception alteration, it is possible that proprioceptive information from other lower limb muscles combined with vestibular cues contributed to improving the state estimates of body sway. This multisensory process likely reduces the overall variance within noisy sensory systems, attenuating body sway [44], which could explain why balance control performance was not poorer during Achilles tendon vibration compared to EVS. However, it is unclear why balance control performance was not better during EVS. Under this condition, unaltered ankle

proprioception could sense body sways, leading to better balance control. One explanation could be that the fusion of ankle proprioception with altered vestibular information led to a noisy unified perception of body sway dynamics and inaccurate state estimation.

When sensory information returned to normal, balance control performance was poorer following Achilles tendon vibration than following EVS. This observation suggests that vestibular reweighting, contrary to ankle proprioception reweighting, could mainly occur at the subcortical level. The vestibular system differs from the proprioceptive system in many ways. First, the same neurons receiving direct afferent inputs can send direct projections to motoneurons, and the first stage of central processing is multimodal [45]. In addition, the vestibular system unambiguously senses head acceleration. Thus, changes in the firing rate of vestibular nerves necessarily provide information about self-motion [46]. By contrast, ankle proprioception either signals the whole-body orientation with respect to the feet or the orientation of the feet with respect to the shin. During ankle dorsiflexion, the brain must determine whether the body sways forward, or the feet are tilted upward. These two situations require balance responses that are fundamentally different, and therefore, such processes should imply complex interactions between cortical and subcortical structures [47–49]. This complex interaction likely causes a slower reweighting. Studies assessing the long-stretch reflexes in lower limbs, due to unexpected surface translation, have reported that these responses are mediated in part by cortical mechanisms [48, 50, 51]. Furthermore, the similarity of the postural responses when different muscles are vibrated means that the motor responses are not caused by the tonic vibration reflex. On the contrary, cortical processing of afferences from all body segments from the feet to the head allows a coherent perception of the whole-body state to be built [52, 53], which is supported by the fact that postural responses, during muscle vibration, are altered by various factors, such as the availability of other sensory cues or balance stability [54–58]. Thus, we suggest that when ankle proprioception returns to normal, the brain must assess the reliability of proprioception, primarily at the cortical level. During the processing of accurate cues, the sum of sensory weight could be transiently larger than one [59]. The slow adjustment of the ankle proprioception weight likely led to an improper corrective ankle torque and therefore a larger center of mass acceleration, which was confirmed by the larger peak force following Achilles tendon vibration offset. Despite the difference between the center of mass acceleration (i.e., RMS value of the scalar distance between the COP and COG) between both single sensory conditions, the similar stabilization time suggests that immediately after Achilles tendon vibration, the balance motor commands were effective in reducing the center of mass acceleration.

### Comparison of the single versus mix sensory conditions

In humans, when the vestibular and proprioceptive systems are simultaneously probed, the amplitude of body sway corresponds to the sum of body sway evoked by the stimulation of the two systems alone [14], and multiple muscle co-vibration does not represent a linear summation of the combined effects [52]. Moreover, neuroimaging studies have found vestibular projections in the primary and secondary somatosensory cortices [15, 16] and the primary motor cortex and premotor cortex [16–18]. Psychophysical studies have revealed that vestibular stimulation facilitates the detection of cutaneous stimuli, suggesting a vestibular-somatosensory perceptual interaction [12]. Thus, there is growing evidence for a functional crossmodal perceptual interaction between vestibular stimulation and the processing of somatosensory inputs. Altering ankle proprioception before stimulating the vestibular system could enhance the sensorimotor mechanisms.

Under the mix sensory conditions, the increase in stabilization time suggests that the sensorimotor mechanisms did not benefit from successive stimulation of different sensory systems. The results of the center of mass acceleration (i.e., RMS value of the COP-COM scalar distance) mitigate the latest affirmation. Nonetheless, it is important to distinguish the RMS value of the COP-COM scalar distance from the stabilization time. The RMS value of the COP-COG scalar distance was calculated over a time window and provided information about the amplitude of the center of mass acceleration. The stabilization time represents a discrete event and provides about the time needed to recover a baseline-like balance control performance. Balance destabilization (i.e., peak force) was a good predictor of the RMS value of the COP-COG scalar distance (variance explained > 80%), while it was not a good predictor of the stabilization time (range of variance explained: 16%–51%). These results suggest that these parameters convey different information about the performance of the sensorimotor control mechanisms.

A limitation of the mix sensory conditions was the delay between sensory stimulations. Previous studies have assessed how ankle proprioception and vestibular information interact with each other when altered simultaneously [14, 60]. The direction and amplitude of body sway during ankle proprioception alterations are influenced by simultaneous changes in the vestibular input. In our study, the aim was to verify whether successive stimulation of two sensory systems involved in balance control could be beneficial for the sensorimotor mechanisms. We added a delay of 5 s between sensory stimulations to avoid a transient effect of the previous stimulation on the following stimulation. Furthermore, the delay needed to be short enough to assure that sensorimotor information was shared over time. Neural responses are improved when different sensory cues are temporally and spatially congruent [61–63]. EVS and Achilles tendon vibration evoked body sways in the same direction. However, it seems that the short delay between sensory stimulation prevented a balance control improvement.

## Conclusion

Understanding how the brain combines sensory information to quickly adapt its motor commands to sudden changes in sensory states represents a challenge. In the present study, when ankle proprioception returned to normal, the peak of ground reaction force was larger, leading to a faster body sway (i.e., a larger RMS value of the COP-COG scalar distance). However, even though balance control mainly relies on ankle proprioception [22–25], balance control performance did not differ during alterations of ankle proprioception and vestibular information. Moreover, successive alterations of different sensory systems (i.e., mix sensory condition) worsen balance control performance as the time needed to recover balance is longer compared to that under the single sensory condition. The amplitude of the center of mass acceleration and the time required to regain balance control seem to convey complementary information about the performance of the sensorimotor control mechanisms.

## Author Contributions

**Conceptualization:** Jean-Philippe Cyr, Noémie Anctil, Martin Simoneau.

**Data curation:** Jean-Philippe Cyr, Noémie Anctil.

**Formal analysis:** Jean-Philippe Cyr, Noémie Anctil, Martin Simoneau.

**Funding acquisition:** Martin Simoneau.

**Methodology:** Jean-Philippe Cyr, Martin Simoneau.

**Project administration:** Martin Simoneau.

**Resources:** Martin Simoneau.

**Software:** Jean-Philippe Cyr, Noémie Anctil, Martin Simoneau.

**Supervision:** Martin Simoneau.

**Writing – original draft:** Jean-Philippe Cyr, Martin Simoneau.

**Writing – review & editing:** Martin Simoneau.

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
