## [Decision Letter · Decision Letter 0]

11 Nov 2019

PONE-D-19-23631

Balance control mechanisms do not benefit from successive stimulation of different sensory systems

PLOS ONE

Dear Dr. Simoneau,

Thank you for submitting your manuscript to PLOS ONE. After careful consideration, we feel that it has merit but does not fully meet PLOS ONE’s publication criteria as it currently stands. Therefore, we invite you to submit a revised version of the manuscript that addresses the points raised during the review process.

We would appreciate receiving your revised manuscript by Dec 26 2019 11:59PM. To enhance the reproducibility of your results, we recommend that if applicable you deposit your laboratory protocols in protocols.io, where a protocol can be assigned its own identifier (DOI) such that it can be cited independently in the future. For instructions see: http://journals.plos.org/plosone/s/submission-guidelines#loc-laboratory-protocols

We look forward to receiving your revised manuscript.

Kind regards,

Bernadette Ann Murphy, PhD

Academic Editor

PLOS ONE

Journal Requirements:

Additional Editor Comments:

As you can see the reviewer only required a couple of small clarifications. I have read the manuscript and I agree with the reviewer. Given the difficulties and delays that we have had in finding a second reviewer and the very positive review of reviewer one, I am willing to go with one review.

Reviewers' comments:

Reviewer's Responses to Questions

**Comments to the Author**

1. Is the manuscript technically sound, and do the data support the conclusions?

Reviewer #1: Yes

2. Has the statistical analysis been performed appropriately and rigorously? 

Reviewer #1: Yes

3. Have the authors made all data underlying the findings in their manuscript fully available?

Reviewer #1: Yes

4. Is the manuscript presented in an intelligible fashion and written in standard English?

Reviewer #1: Yes

5. Review Comments to the Author

Reviewer #1: Overall, a very carefully constructed study, clearly written, and no major revisions recommended.

There are two points for the authors to consider:

1) AP force: what is the rational for this measure. It is assumed that if there is a sustained shift in the sway of the body (forwards, backwards) then the AP GRF is necessary to prevent a fall. This may not be clear to all readers. Therefore, it is recommended that 1-2 sentences be included to better explain what the AP force represents.

2) time after stimulus: there is litterature that suggests that more than 15 seconds of data is required to obtain stable measure of stability (CoP data). Although the study demonstrates that re-stabilization occurrs within a shorter period of time, it does present an opportunity to comment on this if judged relevant by the authors. See the manuscript by Carpenter et al. about minimum time required to obtain stable sample of COP data.

6. PLOS authors have the option to publish the peer review history of their article (what does this mean?). If published, this will include your full peer review and any attached files.

Reviewer #1: No

---

## [Author Response · Author response to Decision Letter 0]

19 Nov 2019

We are grateful to the anonymous reviewer for the productive comments. In the revision, we have tried to address all the criticisms and suggestions as described in the following responses.

Reviewer 1 comment #1:

AP force: what is the rational for this measure. It is assumed that if there is a sustained shift in the sway of the body (forwards, backwards) then the AP GRF is necessary to prevent a fall. This may not be clear to all readers. Therefore, it is recommended that 1-2 sentences be included to better explain what the AP force represents.

Authors reply to comment #1:

As suggested by the reviewer, we added few sentences to clarify the relationship between AP GRF and shift in the AP sway of the body. This can be found in the Methods section of the revised version of the article.

Reviewer 1 comment #2:

Time after stimulus: there is litterature that suggests that more than 15 seconds of data is required to obtain stable measure of stability (CoP data). Although the study demonstrates that re-stabilization occurrs within a shorter period of time, it does present an opportunity to comment on this if judged relevant by the authors. See the manuscript by Carpenter et al. about minimum time required to obtain stable sample of COP data.

Authors reply to comment #2:

We thank the Reviewer for pointing out this reference from Carpenter et al., (2001). We agree with the conclusion of this manuscript. However, in Carpenter et al.’s manuscript, the authors determined how long the sampling duration had to be before observing no more change in the COP measures in absence of alteration in sensory cues. In contrast, in our study, we were interested in assessing the sensorimotor control mechanisms immediately following a sudden change in sensory state. The values for the stabilization time calculated using our algorithm are like the results reported by Asslander and Peterka (2016), as well as Honeine and Schieppati (2014). 

In their manuscript Carpenter et al.’s, assessed how long the sampling duration had to be to observe stable COP measures while in our manuscript, we were interested in determining how long it took to observe a baseline like value of the COP-COG scalar distance. Overall, we think that discussing the difference between our measure of stabilization time and Carpenter et al.’s conclusion would confuse the readers. 

Thank you for considering the revised version of our manuscript,

Sincerely,

Martin Simoneau, PhD

Professor, Laval University, Faculty of Medicine – Department of kinesiology

Researcher, Centre for integrative research and social integration (CIRRIS)

Quebec, Qc, Canada

---

## [Editor Report · Decision Letter 1]

22 Nov 2019

Balance control mechanisms do not benefit from successive stimulation of different sensory systems

PONE-D-19-23631R1

Dear Dr. Simoneau,

We are pleased to inform you that your manuscript has been judged scientifically suitable for publication and will be formally accepted for publication once it complies with all outstanding technical requirements.

With kind regards,

Bernadette Ann Murphy, PhD

Academic Editor

PLOS ONE

Additional Editor Comments (optional):

Nice work and my apologies for all the delays on this one.
---

## [Editor Report · Acceptance letter]

4 Dec 2019

PONE-D-19-23631R1 

Balance control mechanisms do not benefit from successive stimulation of different sensory systems 

Dear Dr. Simoneau:

I am pleased to inform you that your manuscript has been deemed suitable for publication in PLOS ONE. Congratulations! Your manuscript is now with our production department. 

With kind regards,

on behalf of

Dr. Bernadette Ann Murphy 

Academic Editor

PLOS ONE